**Impact of Varying Lidar Measurement and Data Processing Techniques in evaluating**
**Cirrus Cloud and Aerosol Direct Radiative Effects.**
S. Lolli[1,2,1], F. Madonna[1], M. Rosoldi[1], J. R. Campbell[3], E. J Welton[4] J. R. Lewis[2], Y.
Gu[5], G. Pappalardo[1]
[1] CNR-IMAA, Istituto di Metodologie Ambientali Tito Scalo (PZ), Italy
[2] NASA GSFC-JCET, Code 612, 20771 Greenbelt, MD, USA
[3] Naval Research Laboratory, Monterey, CA, USA
[4] NASA GSFC, Code 612, 20771 Greenbelt, MD, USA
[5] UCLA, University of California Los Angeles, Los Angeles, USA
**ABSTRACT**
In the past two decades, ground-based lidar networks have drastically increased in scope
and relevance, thanks primarily to the advent of lidar observations from space and their
need for validation. Lidar observations of aerosol and cloud geometrical, optical and
microphysical atmospheric properties are subsequently used to evaluate their direct
radiative effects on climate. However, the retrievals are strongly dependent on the lidar
instrument measurement technique and subsequent data processing methodologies. In this
paper, we evaluate the discrepancies between the use of Raman and elastic lidar
measurement techniques and corresponding data processing methods for two aerosol layers
in the free troposphere and for two cirrus clouds with different optical depths. Results show
that the different lidar techniques are responsible for discrepancies in the model-derived
direct radiative effects for biomass burning ($0.05$ W/m$^2$ at surface and $0.007$ W/m$^2$ at top
of the atmosphere) and dust aerosol layers ($0.7$ W/m$^2$ at surface and $0.85$ W/m$^2$ at top of
the atmosphere).

[1] Corresponding author: simone.lolli@imaa.cnr.it

Data processing is further responsible for discrepancies in both thin (0.55 W/m$^2$ at surface
and 2.7 W/m$^2$ at top of the atmosphere) and opaque (7.7 W/m$^2$ at surface and 11.8 W/m$^2$
at top of the atmosphere) cirrus clouds. Direct radiative effect discrepancies can be
attributed to the larger variability of the lidar ratio for aerosols (20-150 sr) with respect to
cirrus clouds (20-35 sr). For this reason, the influence of the applied lidar technique plays
a more fundamental role in aerosol monitoring because the lidar ratio must be retrieved
with relatively high accuracy. On the contrary, for cirrus clouds, being the lidar ratio much
less variable, the data processing is critical because smoothing it modifies the aerosol and
cloud vertically resolved extinction profile that is used as input to compute direct radiative
effect calculations.
*1. Introduction*
According to the International Panel for Climate Change (IPCC, 2014), the major
sources of uncertainty relating to current climate studies include direct and indirect
radiative effects caused by anthropogenic and natural aerosols. Further, current estimates
of the global aerosol direct radiative effect remain subject to large relative uncertainties
affecting even the actual sign (indicating either net cooling or heating of the earth-
atmosphere system), which may change from positive to negative diurnally (e.g., Campbell
et al., 2016, Lolli et al., 2017a, Tosca et al., 2017). This depends on the so-called albedo
effect (or the capability of aerosols for reflecting incoming solar light) and whether or not
it is outweighing the greenhouse effect (or the capability of trapping/absorbing outgoing
longwave radiation; Campbell et al., 2016)
Studies on cloud and aerosol optical, geometrical and microphysical properties largely
increased in the last two decades through the abundance of passive ground-based
measurements (i.e., AErosol RObotic NETwork Network; AERONET Holben et al., 1998,
Dubovik et al., 2000, Smirnov et al., 2005, Eck et al., 2014; the Atmospheric Radiation
Measurement program; ARM; Campbell et al., 2002, Ferrare et al., 2006, Perez-Ramirez
et al., 2012, McComiskey et al., 2016; Aerosols, Clouds and Trace gases Research
Infrastructure; ACTRIS Asmi et al., 2013, Pappalardo et al., 2014) or using satellite sensors
(i. e. MODerate resolution Infrared Spectroradiometer; MODIS, Tanré et al., 1997, King
et al., 2003, Remer et al., 2005; i. e. Multi-angle Imaging Spectro-Radiometer; MISR,
Diner et al., 1998, Di Girolamo et al., 2004, Kahn et al., 2009; i.e. Polarization and
Anisotropy of Reflectances for Atmospheric science coupled with Observations from a
Lidar; PARASOL, Tanré et al., 2011; NASA Aerosol-Cloud Ecosystem, ACE, Whiteman
et al., 2018). Nevertheless, these measurements provide only an estimate of the columnar
aerosol (or cirrus cloud) properties.
On the other hand, the Cloud-Aerosol Lidar with Orthogonal Polarization (CALIOP;
Winker et al., 2007), on board of the Cloud-Aerosol Lidar and Infrared Pathfinder Satellite
Observations (CALIPSO) satellite launched by the National Aeronautics and Space
Administration (NASA) in 2006, is capable of estimating range-resolved aerosol and cloud
physical properties. However, the sun-synchronous orbit limits spatial and temporal
coverage (orbital revisit time period of 16 days) that make the datasets difficult to apply
and interpret for specific forms of process study. The vertical structure of cloud and aerosol
properties can also be retrieved through combined lidar and radar ground-based
measurements as proposed in the frame of the CloudNet European Project (Illingworth et
al., 2015). Still, the radar technique proves capable of characterizing only the relatively
extreme fraction of the aerosol size distribution (Madonna et al., 2010, Madonna et al.,

70 2013).

Based on the progress in optical technologies in the late 1990's and the beginning of
2000's, federated ground networks of lidars were established [NASA Micro Pulse Lidar
NETwork(MPLNET), Campbell et al., 2002, Welton et al., 2002, Lolli et al., 2013;
European Aerosol Research LIdar NETwork, (EARLINET) Pappalardo et al., 2014, Asian
Dust NETwork (ADNET), Sugimoto et al., 2010, Latin American Lidar NETwork
(LALINET), Antuña-Marrero et al., 2015, Lolli et al., 2015], the bulk of which are based
on single or dual-channel elastic and Raman lidar instruments. The Eulerian viewpoint of
ground-based lidars is providing important contextual measurements relative to satellite
profiling, like from CALIOP (Winker et al., 2007).
The emerging prominence of ground-based lidar, however, strengthens the necessity
for further studies of optical, geometrical and microphysical aerosols and clouds properties
resolved from multi-spectral lidar techniques, as claimed by several papers (Pappalardo et
al., 2004, Mona et al., 2006, Wang et al., 2012, Pani et al., 2016, Lolli et al., 2013, Campbell
et al., 2016, Lolli et al., 2017). Multi-spectral and Raman lidars can retrieve aerosol and
cloud properties with much better accuracy than elastic lidars, without many fundamental
assumptions, (e.g. Ansmann et al., 1992; Goldsmith et al. 1998, Mona et al., 2012,
Pappalardo et al., 2014), thought with greater operational expenses. The High Spectral
Resolution Lidar (HSRL - Shipley et al., 1983; Grund and Eloranta, 1991) technique allows
for the separation of molecular and aerosol signals, and thus affords an independent
retrieval of aerosols extinction and backscattering coefficients. However, the technology
remains relatively complex and expensive, making them an unattractive choice for
operational networks (e.g. Hair et al., 2008).
The Raman technique (section 2.2) permits retrieval of aerosol and cloud vertically-
resolved extinction coefficient without any binding assumptions, which are the cornerstone
of elastic-based retrieval techniques (section 2.1). Certain instabilities exist, however
(Ansmann et al., 1992, Wandinger et al, 1995). In order to reduce the random uncertainty
affecting the retrieval, a smoothing of the range-resolved profile is required at expense of
the effective vertical resolution (Pappalardo et al., 2004, Iarlori et al., 2015) of the
extinction coefficient profile.
Ultimately, different lidar techniques and/or processing algorithms lead to
differences of the retrieved vertically-resolved particulate optical properties, affecting the
apparent significance, position and the geometry of observed aerosol and cloud layers. The
impact of these differences has never been extensively evaluated. Since lidar-derived
optical properties obtained from different instrument techniques are more and more
frequently used to assess the direct radiative effects of clouds and aerosols (e.g., Campbell
et al., 2016, Lolli et al., 2017a, Tosca et al., 2017), corresponding uncertainties in
determining direct radiative effects, which may help reconcile inconsistencies in studies
carried out at the global scale based on different lidar techniques, are compulsory,
especially now that several new space missions with lidar on board have been launched
(Cloud-Aerosol Transport System; CATS, McGill et al., 2015) or are scheduled (European
Space Agency Earth Care mission; Illingworth et al., 2015).

112   The primary goal of this paper is to evaluate the relative differences between the

aerosol/cloud direct radiative effects both at surface (SFC) and at the top-of the-atmosphere
(TOA) computed using the aerosol/cloud optical properties estimated from more
sophisticated versus basic lidar techniques (i.e., Raman vs. elastic lidar). To reach this goal,
we use the Fu-Liou-Gu (FLG; Fu and Liou, 1992, Fu and Liou, 1993, Gu et al., 2003, Gu
et al., 2011, Lolli et al, 2017b) radiative transfer model to calculate the difference in net
direct radiative effect for aerosols and clouds at TOA and SFC for profiles derived from
both elastic and combined Raman/elastic lidar techniques.

**_2. Method_**
_2.1 Elastic and Raman Lidar techniques_

123   Elastic-scattering lidar instruments require assumptions and careful consideration of

measurement strategies to constrain the single-scattering lidar equation (Eq. 1), defined as
$$P_r(r) = O(r)K\frac{\beta(r)}{r^2}exp^{-2\int_0^r \alpha(r')dr'}$$

$,$     (1)
where $P_r(r)$ is the received power at a range $r$ *and* $O(r)$ is the overlap function, which
depends on intersection between the respective telescope and laser field of view. *O(r)*
equals to unity for a distance $r_0$ depending on the specific lidar system, spanning from few
hundred meters to 4-5 km for Micro Pulse Lidar systems (MPL; Campbell et al., 2002). *K*
is the so-called lidar constant (instrument dependent, function of detector quantum and
optical efficiencies, telescope diameter, etc.), followed by the two unknown variables, *β(r)*
the total backscattering coefficient and *α(r)* the total extinction coefficient.
A classical method to solving Eq. (1) for single-channel elastic-backscatter lidars
(Fernald, 1984) is based on the assumption of the columnar-averaged value of the ratio
between the two unknown coefficients, typically indicated by *S* and called "lidar ratio".
The method, due to the large variability of *S* (i.e., 20-150 sr for aerosols; Ackermann, 1998;
Ferrare et al., 2001, Sakai et al., 2003, Müller et al., 2007, Groβ et al., 2011, 2013 and
2015, Veselovskii et al., 2015) translates into large uncertainties associated with the
retrieval of *α* and *β* (Lolli et al., 2013).
Through greater spectral complexity, it is possible to retrieve *α* and *β* with multi-
spectral lidars without relying too heavily on fundamental assumptions. For instance, the
combined detection of the elastic-backscattered and inelastic backscattered radiation due
to the Raman effect by nitrogen (or oxygen) molecules excited to a different vibrational or
rotational energy level is possible. Using the Raman lidar technique, we can constrain and
rewrite Eq. (1) as

$$\alpha_{\lambda_L}^{par}(r) = \frac{\frac{d}{dr}\left\{\ln\left[n_R(r)/P_r(r)r^2\right]\right\} - \alpha_{\lambda_L}^{mol}(r) - \alpha_{\lambda_R}^{mol}(r)}{1 + \left(\frac{\lambda_L}{\lambda_R}\right)^{\mathring{a}}}$$
(2)

where $\lambda_L$ is the elastic wavelength while $\lambda_R$ is the wavelength of the Raman scattering,
$\alpha_{\lambda_L}^{par}(r)$ represents the particle (aerosols or clouds) extinction coefficient at elastic
wavelength at range $r$ while $\alpha^{mol}_{\lambda_L}(r)$ and $\alpha^{mol}_{\lambda_R}(r)$ are the molecular extinction coefficients
at wavelengths $\lambda_L$ and $\lambda_R$ respectively, $P_r(r)r^2$ is the detected range corrected Raman
signal from range $r$, while $n_R(r)$ represents the number density of range-resolved scatters.
The wavelength dependence of the particle extinction coefficient is described by the
Ångström coefficient, $\mathring{a}$, defined from the relation
$$\qquad \frac{\alpha^{par}_{\lambda_L}(r)}{\alpha^{par}_{\lambda_R}(r)} = \left(\frac{\lambda_R}{\lambda_L}\right)^{\mathring{a}} \qquad\qquad (3)$$
Eq. (2) allows for independently retrieving vertically-resolved optical coefficients with
only very limited *a-priori* assumptions (the Ångström coefficient should be estimated or
assumed, but this estimate or assumption, involving a ratio, typically amounts to less than
5% of total error; Ansmann and Müller, 2005). The particle backscattering coefficient,
$\beta^{par}_{\lambda_R}(r)$ and $\beta^{par}_{\lambda_L}(r)$ , can be derived directly from the ratio of the Raman signal at $\lambda_R$
and the elastic signal at $\lambda_L$.

*2.2 Fu-Liou-Gu Radiative Transfer Model*
To calculate aerosol and cloud direct radiative effects, we use the one-dimensional FLG
radiative transfer model, developed in the early 1990's. The original code has been adapted
to retrieve cloud and aerosol direct radiative effects using the aerosol and cloud vertical
profile of lidar extinction as input. There exist several parameterizations that provide the
vertical profile of cloud microphysics using lidar-retrieved cloud extinction profile, each
one with pros and cons, as showed in Comstock et al. (2007). For the purpose of this study
and also considering authors past experience (Campbell et al., 2016, Lolli et al., 2017a),
we parameterize cirrus clouds through the Heymsfield et al., (2014) empirical relationship
conceived expressly for lidar measurements. Here, the cirrus cloud ice crystal average
diameter is directly proportional to the absolute atmospheric temperature (obtained through
a radiosonde, regularly launched at measurement site, or numerical reanalysis dataset).
Cirrus cloud optical depth and crystal size profiles are used to calculate the single scattering
albedo (SSA), phase function and asymmetry factor (AF) at each level.

Similarly, FLG calculates the direct radiative effect of aerosols as a function of the

partial contribution of each aerosol species to the total optical depth at each altitude level.
FLG uses a lookup table (LUT) with single scattering properties for eighteen different types
of aerosols coming from the OPAC (Optical Properties of Aerosol and Clouds) database
(d'Almeida et al., 1991; Tegen and Lacis, 1996; Hess et al., 1998). Among all aerosol
species, for the initial cases introduced in Section 2.2 we assume that the dust layer is
constituted by pure dust advected from Saharan region (aerosol type 17 in FLG), while in
the second case we assume pure biomass burning aerosol (aerosol type 11 in FLG).
Nevertheless, if the measured aerosol atmospheric profiles do not match exactly the two-
selected aerosol types this does not affect the results because we are interested in evaluating
the relative discrepancies among the different lidar techniques/data processing. Therefore,
what is most relevant in the approach is the application of the same parameterization to
each of the different techniques/data processing.

The aerosol/cloud direct radiative effect is calculated subtracting from the FLG total

sky run (where aerosols or clouds are present) the FLG run with a pristine atmosphere
(control), expressed as
$$DRE = FLG^{TotalSky} - FLG^{Pristine} \qquad , \qquad (4)$$
where $DRE$ is the direct radiative effect (from aerosols or clouds), while the superscript
*TotalSky* means that FLG is computed taking into account the aerosol/cloud profile and
*Pristine* represents a hypothetical "clear-sky" atmosphere with no aerosols or clouds.

Direct measurements of aerosol microphysical properties require multi-wavelength

lidar (e.g. Veselovskii et al., 2002, 2013), which are not common in many networks and
also are sensitive to systematic and random errors in the optical data (Perez-Ramirez et al.,
2013). We focus here on lidar systems that can operate continuously in different networks,
and our direct radiative effect calculations do not vary much when changing effective
radius and single scattering albedo.

*2.3 Direct radiative effect computation*
For the analysis in this study, we analyzed lidar data collected with the MUlti-
wavelength System for Aerosols (MUSA) Lidar (Madonna et al., 2011), deployed at
Consiglio Nazionale delle Ricerche (CNR), Istituto di Metodologie per l'Analisi
Ambientale (IMAA) Atmospheric Observatory (CIAO) in Potenza, Italy (40.60N, 15.72E,
760m above sea level; a.s.l). MUSA is a mobile multi-wavelength lidar system based on a
Nd:YAG laser source equipped with second and third harmonic generators and on a
Cassegrain telescope with a primary mirror of 300mm diameter.
MUSA full angle field-of-view (FOV) and laser beam divergence are large enough (1.0
mrad and 0.6 mrad, respectively) to add important multiple scattering (MS) contributions
to the retrieved cirrus extinction coefficient profiles. The Raman extinction coefficient
profiles have been corrected for MS as described in Wandinger (1998), taking into account
MS contributions by introducing in the respective lidar equation the multiple scattering
parameters. These parameters have been calculated, by applying Eloranta's model
(Eloranta, 1998) to estimate the contributions of individual orders of multiple scattering.
In the model simulations, MUSA specifications (FOV and laser beam divergence) have
been used, and a mono-disperse size distribution profile of cirrus cloud ice crystals has
been assumed with effective diameters derived from the same parameterization used in
FLG model (Heymsfield et al., 2014). The first five scattering orders have been summed.

MUSA lidar system is not tilted due to technical constraints. However, the averaged
cirrus cloud retrieved lidar ratios from the combination of Raman and elastic lidar
techniques (corrected for MS effects) are 24sr and 26sr, for cirrus cloud cases highlighted
here from 10 June 2010 and 17 February 2014, respectively. Those values are consistent
with a very low probability of significant specular reflection. The previous statement is
supported by the fact that crystal size diameter computed with Heymsfield et al. (2014)
parameterization is below 100μm, a threshold value above which the specular reflection
can arise. Moreover, in Hogan and Illingworth (2003) work, it is founded that specular
reflection tends to be much stronger and more common for temperatures between 250 K
and 264 K (that corresponds to much lower altitudes with respect to the examined cirrus
cloud cases), where plate crystals, which induce the greatest specular signal, are most
common.
The three laser beams at 1064, 532 and 355nm are simultaneously and coaxially
transmitted into the atmosphere in a biaxial configuration. The receiving system has three
channels for the detection of the radiation elastically backscattered from the atmosphere
and two channels for the detection of the Raman radiation backscattered by the atmospheric
$N_2$ molecules at 607 and 387 nm. The elastic channel at 532 nm is split into parallel and
perpendicular polarization components by means of a polarizer beamsplitter cube. The
backscattered radiation at all the wavelengths is acquired both in analog and photon
counting mode. The typical vertical resolution of the raw profiles is 3.75 m with a temporal
resolution of 1 min. The system is compact and transportable. It has operated since 2009,
and it is one of the reference systems used for the intercomparison of lidar systems within
the EARLINET (Pappalardo et al., 2014; Wandinger et al., 2016) Quality Assurance
program. In this paper, the data analysis has been carried out considering four observation
scenarios at night, as the Raman channel signal shows a much higher signal-to-noise ratio
during nighttime:
1) **Dense Dust Aerosol and Biomass Burning Events**. The aerosol extinction
profiles are retrieved using the UV (355nm) channel. For each case, the extinction
profile is retrieved both with the Raman technique (Ansmann et al., 1990,
Whiteman et al., 1992, Veselovskii et al., 2015) and estimated using the sole elastic
channel, applying an iterative algorithm (Di Girolamo et al., 1999) with an assigned
lidar ratio (S=57 sr for dust case, Mona et al., 2006 and S=63 sr for biomass
burning, retrieved averaging the lidar ratio from MUSA Raman channel). Both the
Raman and elastic lidar signals have been smoothed by performing a binning of 16
range gates, resulting in a vertical resolution of 60 m. For the Raman channel
retrieval, the extinction profile has been calculated using the sliding linear fit
technique, with a bin number resulting in an effective vertical resolution of 360 m
(Pappalardo et al., 2004). For the elastic channel retrieval, the estimated extinction
profile has been first calculated with the signal full vertical resolution of 60 m and
then smoothed to the same effective vertical resolution as the Raman extinction
profile (360m), using a $2^{nd}$ order Savitzky-Golay smoothing filter (Press et al.,
1992; Iarlori et al., 2015).

2) **Thin and Opaque Cirrus Clouds**. Like aerosols, cirrus cloud extinction profiles
are retrieved using the UV (355nm) channel with the Raman technique. The elastic
channel retrieval for thin cirrus cloud is obtained applying the same iterative
algorithm followed for dust and biomass burning. Although, for the opaque cirrus
cloud, due to convergence problems of the iterative method for higher cloud optical
depths, we used the MPLNET Level 1.5 cloud product algorithm (Lewis et al.,
2016) based on a Klett inversion (Klett, 1985). For both cases (iterative and
MPLNET), we assumed a fixed lidar ratio value obtained from Raman and elastic
measurements corrected by MS effects of 24sr for thick and 26sr for thin cirrus
cloud.

The Raman extinction profile has been calculated with an effective vertical

resolution of 420 m (thin cirrus cloud) and 780 m (opaque cirrus cloud),

respectively. The iterative (thin cirrus) and MPLNET Level 1.5 cloud algorithm

(opaque cirrus; Lewis et al., 2016) extinction profiles are calculated with the

original signal vertical resolution of 60 m and smoothed at a resolution of 420 m

(thin cirrus) and 780 m (opaque cirrus), respectively, using the Savitzky-Golay

filter to match Raman channel spatial resolution.

3) The thermodynamic profile of the atmosphere, needed to calculate the direct

radiative effect, is estimated using a standard thermodynamic profile (USS976)

mid-latitude model. Emissivity and albedo values are taken from the MODIS

Bidirectional Reflectance Distribution Function (BRDF)/Albedo algorithm product

(Strahler et al., 1999), with a spatial resolution of 0.1 degrees averaged over a 16-

289          day temporal window (Campbell et al., 2016). As each measured cloud and aerosol

extinction profile comes with a relative uncertainty per range bin, the sensitivity of

FLG to the input parameters is evaluated applying a Monte Carlo technique. Each

extinction profile is replicated 30 times (i.e. a number statistically meaningful),

running the MonteCarlo code on the original profile random uncertainty. Likewise,

for each replicated extinction profile, the Monte Carlo technique gives a value of

surface albedo and profile temperature, based on their respective uncertainties. The

direct radiative effect parameters derived for each profile are then represented with

a boxplot. It is possible then to quantify the effect of the smoothing calculating the

uncertainty from the mean and the standard deviation of the values of net forcing.


**3. Results**
*3.1 Dust and Biomass Burning Event*
The analyzed dust event is retrieved from measurements taken on 03 July 2014 at
CIAO. Figure 1 shows both the range-corrected composite signal at 1064 nm (Fig. 1a, left
panel), and the lidar aerosol extinction profiles at 355 nm (Fig. 1b, left panel) obtained
using the Raman technique with an effective resolution of 360 m and estimated using the
elastic lidar technique at two different resolutions (60 m and 360 m) and a fixed $S$ value
obtained analyzing climatological data (S=57sr; Mona et al., 2006). The Raman extinction
profile is noisier with respect to those obtained with the iterative method. All profiles,
calculated with an integration time of 121 minutes, in the time window from 19:34 UT to
21:40 UT show no significant aerosol loading above 5.5 km.
Figure 3a shows the difference between the estimation of the direct radiative effect
using the two considered lidar techniques and data processing at the top-of-the-atmosphere
(TOA; Fig 3a, left panel) and surface (SFC; Fig. 3a right panel). The most important
contribution to this difference in FLG calculations for this case is related to the adopted
lidar technique (red arrows in Fig. 3a, left and right panels) and not to the effective vertical
resolution determined by the smoothing (blue arrows in Fig. 3a, left and right panels). This
characteristic is invariant switching from TOA (Fig. 3a right panel) to SFC (Fig. 3a left
panel) and is mainly the result of the assumption of a fixed lidar ratio to estimate the aerosol
extinction profile using the elastic technique.
For the dust case, the net direct radiative effect determined with the two different lidar
techniques differs by 0.7 W/m$^2$ (5%) at SFC and 0.85 W/m$^2$ (6%) at TOA. In absolute
magnitudes, these net total forcing values are larger than the uncertainty, on average,
estimated direct effect by IPCC (mean -0.5 W/m$^2$, range -0.9 to -0.1). The contribution due
to smoothing is negligible in comparison.
The analyzed biomass burning case study is retrieved from measurements taken on 19
June 2013 at CIAO integrating the signal temporally from 19:27 UT to 20:48 UT. The
extinction profiles used as input into the FLG radiative transfer model were retrieved in the
same way as for the dust case. Instead of a climatological lidar ratio value at 355nm,
however, we used $S$=63 sr, obtained by averaging the lidar ratio profile retrieved with
combined Raman-elastic techniques in the biomass burning layer. In Figure 1b (right
panel) are the extinction profiles obtained from both the Raman and iterative methods (full
resolution and smoothed over 360m window). Figure 3b shows the difference in biomass
burning direct radiative effects with respect to the different lidar and data processing
techniques. Similar to the dust case event, the bigger differences are found to be related to
the different lidar techniques both at SFC (0.05 $W/m^2$ or 5%; red arrows, Fig. 3b right
panel) and at TOA (0.007 $W/m^2$ or 5%; Fig. 3b left panel).
The analysis shows how the mixing of different lidar techniques in a specific study or
in the routine operations of an aerosol network at regional or global scale must take into
account of the uncertainties related to the assumptions that are behind the retrieval of the
optical properties. This is important not only to provide a complete assessment of the total
uncertainty budget for each lidar product, but also to enable a physically consistent use of
the lidar data in the estimation of the direct radiative effect and, likely, for many other user-
oriented applications based on lidar data.

*3.2 Cirrus cloud*
Similar to Fig.1, Figs. 2a and 2b shows the composite range-corrected signal and three
extinction profiles retrieved from Raman lidar measurements of cirrus clouds with a
vertical resolution of 420 m (thin cirrus, Fig 2a,b left panel) and 780 m (opaque cirrus, Fig
2a,b right panel), and with the elastic channel at two vertical resolutions (60m and 420m
iterative method for thin cirrus cloud; 60m and 780 m MPLNET Level 1.5 cloud product
algorithm for opaque cirrus cloud) using a MS corrected lidar ratio of 24sr (opaque cirrus)
and 26sr (thin cirrus) . The obtained cloud extinction profiles from the different lidar and
data processing techniques are averaged over 42 minutes, in the time window from 01:29
UT to 02:13 UT on 17 February 2014 (thin cirrus) and from 19:40 UT to 20:44 UT on 10
June 2010 (opaque cirrus), respectively.
Figure 4a depicts the results obtained for cirrus cloud measurements taken on 17
February 2014. Here we have a completely different situation with respect to the aerosol
cases. That is, the discrepancies between the Raman and elastic lidar techniques (red arrows
in Fig. 4a, left and right panels) are much smaller than the discrepancies due to the effective
vertical resolution of the extinction coefficient profile both at TOA and SFC (blue arrows
in Fig. 4a, left and right panels). This is related to what is typically a much stronger
extinction coefficient for clouds than for aerosols. In this cirrus cloud case, the direct
radiative effect determined with the two different lidar techniques differs by about 1.2
$W/m^2$ (16%) at TOA and 0.04 $W/m^2$ (4%) at SFC, while the effect of smoothing within a
window of 420 m provides an additional difference of 2.7 $W/m^2$ (47%) at TOA and about
0.55 $W/m^2$ (53%) at SFC.
Results from the opaque cirrus cloud (Fig. 4b, left and right panels) exhibit a similar
behavior to the thin cirrus cloud, with signal smoothing outweighing the impact of the lidar
technique (blue arrow). The order of magnitude is similar to the thin cirrus cloud, with a
difference at TOA between techniques of 4.6 $W/m^2$ (14%) and 1.6 $W/m^2$ (11%) at SFC. In
contrast, the difference in data processing is of 11.8 $W/m^2$ (39%) at TOA and 7.7 $W/m^2$
(64%) at SFC. The results are evidence of the critical need to study cirrus clouds using
high-resolution profiles of the optical properties to provide an accurate estimation of the
cloud direct radiative effect.

*4. Conclusions and future perspectives*
We applied the adapted Fu-Liou-Gu (FLG) radiative transfer model to quantitatively
evaluate how much the lidar and/or data processing technique applied influence the net
direct radiative effect exerted by two different upper atmospheric aerosol layers (dust and
biomass burning) and a thin versus opaque cirrus cloud layer, both at top-of-the-
atmosphere (TOA) and surface (SFC). The evaluation has been made using aerosol/cloud
extinction atmospheric profiles as inputs into FLG radiative transfer model retrieved using
the Raman/elastic technique and as estimated by lidar elastic measurements only (iterative
method for aerosol layers and thin cirrus cloud; NASA Micro-Pulse Lidar Network Level
1.5 cloud algorithm for opaque cirrus cloud). Because the Raman measurement retrieval is
unstable due to the derivative of the signal at the numerator (see Eq. 2), a smoothing of the
range-corrected signal is necessary to reduce the associated random uncertainty. The same
processing treatment has been applied also to the elastic measurement signals.

The results show that the difference in direct radiative effect between the lidar and data

processing/smoothing techniques applied is mostly unvaried at TOA and SFC. For the dust
and biomass burning episodes, the data processing/smoothing does not play a major role,
but instead the lidar measurement technique is more important with respect to the final
result. This can be explained by the large variability of the lidar ratio (i.e., the unknown
extinction-to-backscatter ratio used to constrain the single-solution lidar equation)
compared to the assumed value. The opposite is true for cirrus clouds, where the applied
data processing/smoothing play a fundamental role in determining sensitivities in the final
results. This is due to the smoothing effect on the observed sharp structures that strongly
alters the vertical structure and the extinction of the cloud.

Summarizing, we found that for the aerosol cases, the main difference both at TOA

and SFC is driven by the respective lidar technique and not the data processing, with a
difference on dust direct radiative effect of 0.7 $W/m^2$ (5%) at SFC and 0.85 $W/m^2$ (6%) at
TOA. Similarly, for biomass burning we found a discrepancy 0.05 $W/m^2$ (5%) at SFC and
0.007 $W/m^2$ (5%) at TOA. For the cirrus clouds, the data smoothing is producing larger
differences with respect to the lidar technique. On the contrary, using a different data
processing/smoothing implies a larger difference in cirrus cloud direct radiative effect. A
discrepancy of 0.55 W/m$^2$ (53%) is found at SFC while about 2.7 W/m$^2$ (47%) at TOA for
the thin cirrus cloud. Similarly for the opaque cirrus, the discrepancies produced by data
processing/smoothing are larger with respect to the different lidar technique. At SFC we
find a difference of 7.7 W/m$^2$ (64%) and 11.8 W/m$^2$ at TOA (39%).
A possible explanation of this different behavior is that the FLG radiative transfer
model calculations are strongly dependent on the optical depth of the examined
atmospheric layer. At coarse resolution (cloud) the smoothing is producing changes in the
extinction profile that translates into creation/suppression of ice crystals that have a strong
influence on direct radiative effect. At finer resolution, as in the case of aerosol case studies,
the smoothing is just producing fluctuations that do not influence the total radiative effect.
In this case, the lidar technique is making a big difference, as an assumed wrong value for
lidar ratio ($S$) that has a much larger variability with respect to the clouds, will amplify or
suppress the aerosol peak that will translate into a higher/lower radiative effect.
With this study, we wish to draw attention in speculating how much the derived aerosol
and cloud radiative effect is dependent on the lidar measurement and retrieval techniques,
as well as on the data processing constraints/assumptions. This dependence looks
increasingly relevant for existing and future space missions involving lidar instrument, as
well as for the GAW Atmospheric LIdar Observation Network (GALION; Hoff et al.,
2008) project, which features then main objective of federating all existing ground-based
lidar networks to provide atmospheric measurement profiles of the aerosol and cloud
optical and microphysical properties with sufficient coverage, accuracy and resolution. For
future work, it is imperative on the community to continue understanding and refining what
are the limits of the each lidar technique along with the related retrieval algorithms adopted
in each ground-based network. FLG or any other well-established radiative transfer model
then can be used as diagnostic tool to assure data quality through continued
intercomparisons with real observation both at ground (using flux measurements), in situ
(aircraft measurements) and at TOA (using satellite-based measurements).

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

*Research: Atmospheres*, *107*(D19).


*Figures*

**a)**

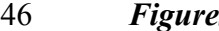

**b)**

Figure 1 a): composite plot of the range corrected signal at 1064nm showing a well-defined dust layer

at about 5 km a.s.l. (left panel) and for a biomass burning aerosol layer at about 2 km (right panel). b):

aerosol lidar extinction profiles at 355nm retrieved with the Raman and the elastic lidar techniques with

different spatial resolutions (60m and 360m) for dust (signal temporally integrated from 19:34UT to

21:40UT) outbreak on 3 July 2014 (left panel) and for biomass burning (signal temporally integrated

from 19:27UT to 20:48UT) on 19 June 2013 (right panel). The iterative method used a fixed lidar ratio

value of S=45sr, determined by climatological measurements (Mona et al., 2006) for the dust aerosol

layer. For the biomass burning we used the averaged value of S=63sr obtained from MUSA Raman lidar.

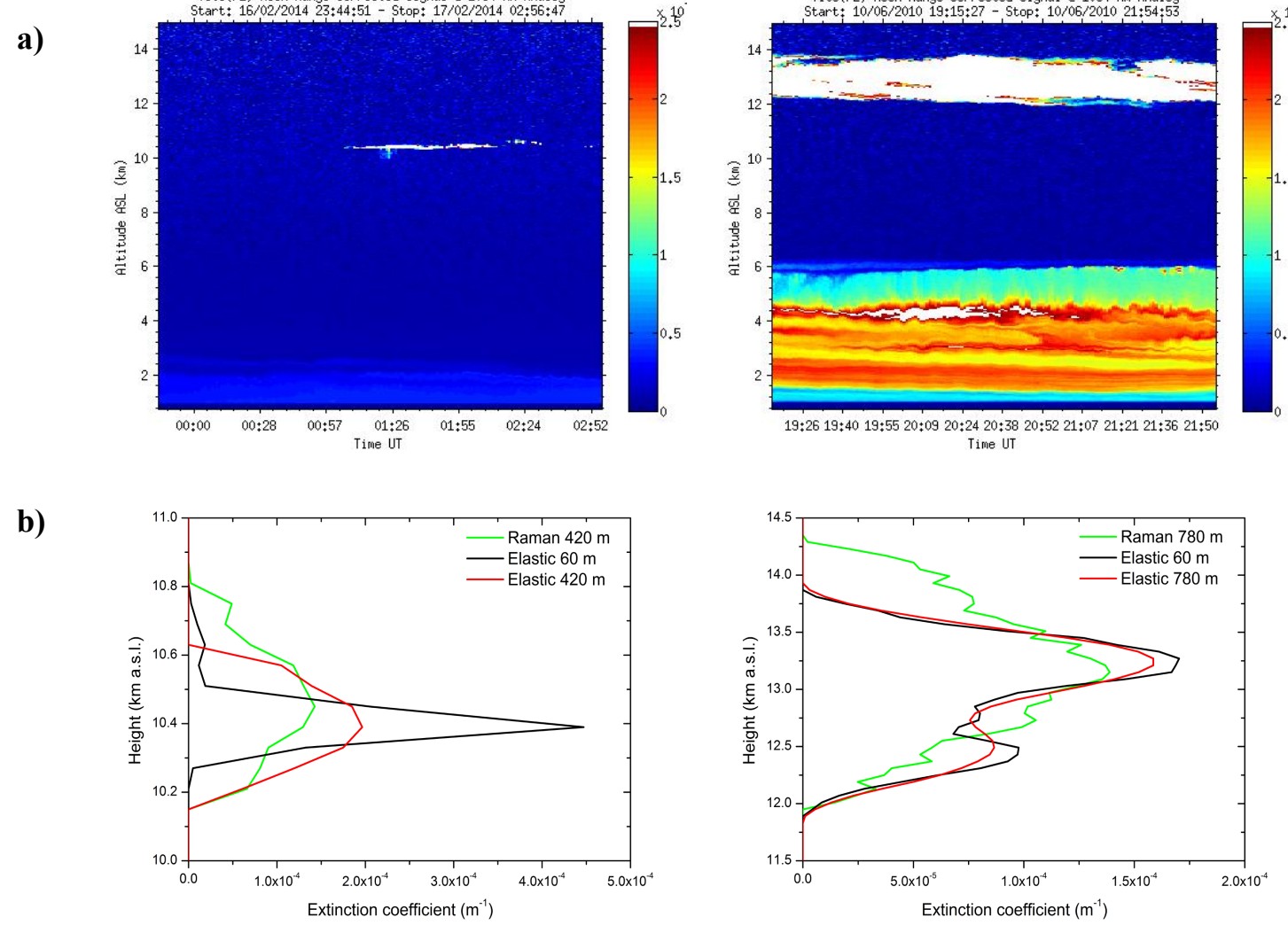

Figure 2: a) composite plot of the range corrected signal at 1064nm showing a thin cirrus cloud at about 10km (right panel) and an opaque cirrus cloud at about 12.5 km. b) left panel: lidar extinction profiles at 355nm from Raman and elastic channel respectively a cirrus cloud on 17 February 2014 (signal temporally integrated from 01:29UT to 02:13UT). The iterative method at the two different resolutions (60m and 420m) used a fixed S value (*25sr*), determined by climatological measurement. Figure 2a, b) right panels: same as Figure 2a, b) left panels but for a cirrus cloud detected on 10 June 2016 (signal temporally integrated from 19:42UT to 20:44UT). The Raman lidar channel is smoothed over a 420m and 780m spatial window.  On 10 June 2016, the elastic channel is retrieved using MPLNET algorithm (Lewis et al., 2016) with *S=25sr* at 60m and 780m respectively.

770

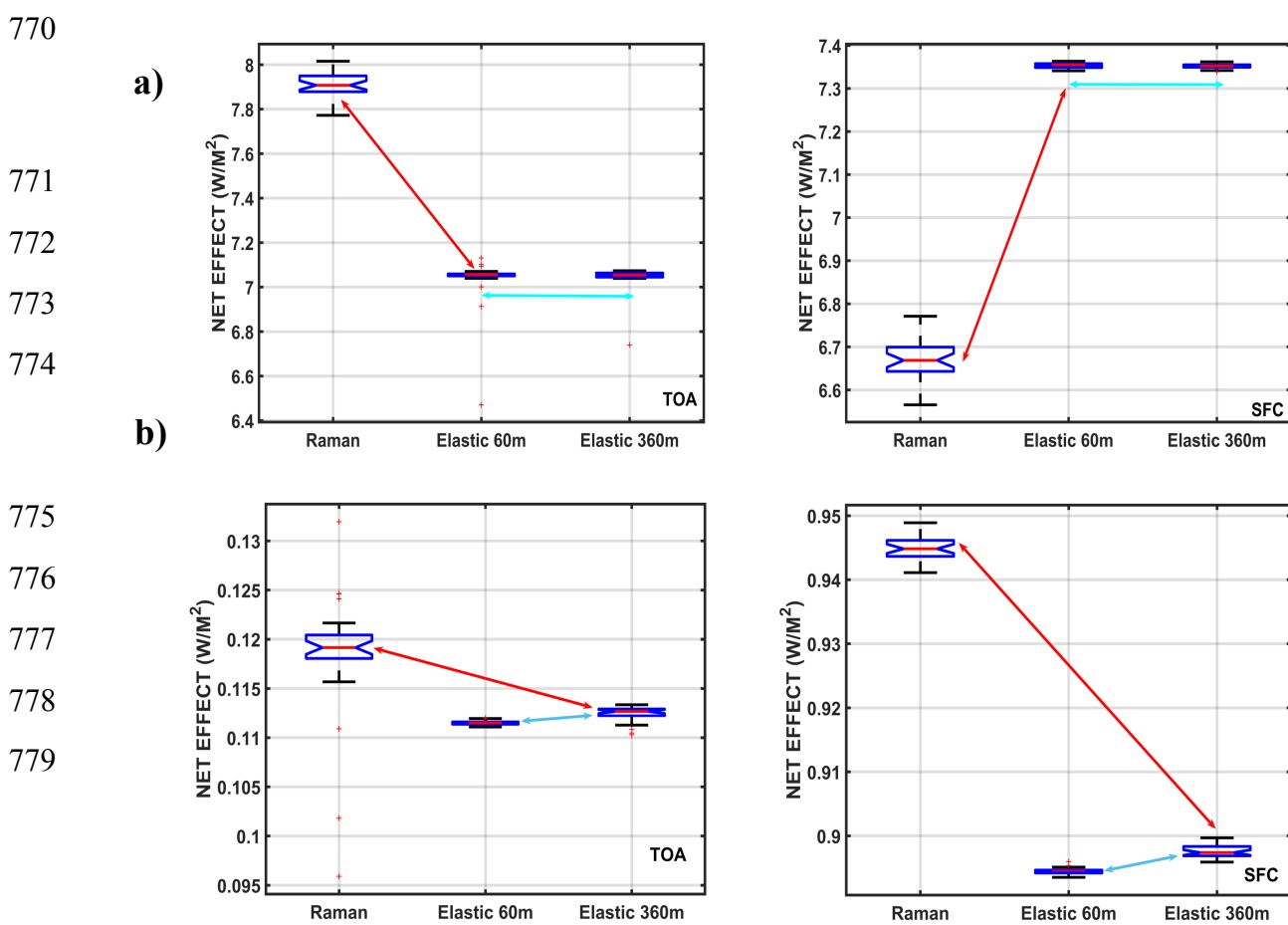

Figure 3. The direct radiative effect, for the dust aerosol case study (Figure 3a) on 03 July 2014 and biomass
burning case on 19 June 2013 (Figure 3b) computed for retrievals obtained with Raman lidar channel
smoothed over a window of 360m, elastic channel at full resolution (60m) and elastic channel smoothed over
a 360m window to be compared with Raman channel . The results are represented as a distribution of values
obtained with the MonteCarlo simulations by the boxplots, is calculated at TOA (left panel) and SFC (right
panel) respectively. As it is clearly visible, the larger discrepancy in forcing is related mostly to the lidar
measurements technique (red arrows), not on the data processing constraints/assumptions (blue arrows).






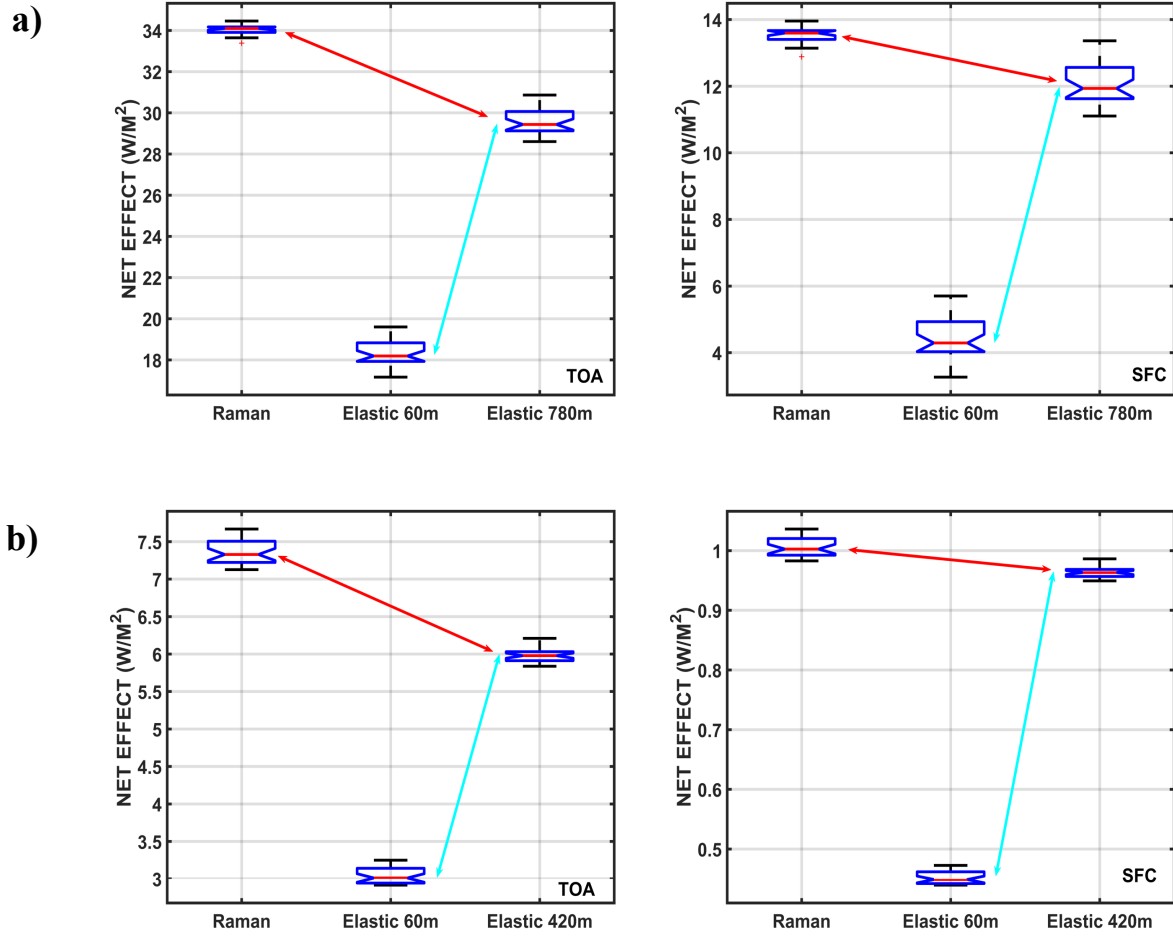

Figure 4 Same as Figure 3 but for two cirrus cloud cases (Fig. 4a, 17 Feb 2014, Fig 4b, 10 June 2016).
The Raman lidar channel is smoothed over 420m window for cirrus on 17 Feb. 2014 and 780m window
for cirrus on 10 June 2010. The net radiative effect is calculated at TOA (left panel) and SFC (right
panel) respectively. As it is clearly visible, in both cases the larger discrepancy in radiative effect is
related mostly to the data processing (blue arrows), not on lidar technique (red arrows).