# Peer review of "Impact of Varying Lidar Measurement and Data Processing Techniques in evaluating 2 Cirrus Cloud and Aerosol Direct Radiative Effects. S. Lolli1,2,1, F. Madonna1, M. Rosoldi1, J. R. Campbell3, E. J Welton4 J. R. Lewis2, Y. 3 4 Gu<sup"

_Atmospheric Measurement Techniques, 2017_

## Referee Comment (RC1) · Anonymous Referee #1 · 5 Jul 2017

General

The paper deals with radiative transfer estimations based on lidar profile observations of dust and cirrus. The goal is to show the impact of the lidar retrieval method.

I have a problem with the methodology (see details).

Major revisions are needed.

[Figure]

Details

Although the paper deals with lidar observations of cirrus extinction profiles, there is no information on the laser beam pointing (zenith or off zenith to avoid specular reflection) and no information about the receiver field of view which has an impact on the multiple scattering contribution. On the other side, the depolarization technique is explained (even the 45 deg calibration) although not used.

Please re-write this section, update the instrument part to meet the requirements for this paper.

Now, I come to my most important point:

The authors use both, the Raman lidar method and the Klett retrieval to determine particle extinction profiles. And EARLINET members (experts in the field of Raman lidars) probably know that the optimum Klett solutions of the backscatter and the extinction profiles are obtained with the 'actual' lidar ratio (profile) from the Raman lidar observations. Ideally, Klett and Raman backscatter and extinction profiles coincide, ... but usually the available Klett codes cannot handle lidar ratio profiles.

However, if you apply the method to such a rather rather thin cirrus as done in this paper, then we may have a problem. I would recommend to use a visible, very well developed cirrus cloud deck (not this subvisible cirrus with an optical depth 0f about 0.02). Is there a reason why this quite unusual cirrus is taken, and not a very normal one?

Nevertheless, by just taking a climatological value for the dust lidar ratio of 45 sr and for the cirrus of 25 sr in the Klett retrievals, and in this way by completely ignoring the reality, i.e., the 'actual' Raman lidar observations of the lidar ratio . . .. it is not surprizing that you obtain different Klett and Raman extinction profiles. The true ones are, by the way, the Raman solutions. The Klett solutions are wrong.

If your Klett code cannot handle lidar ratio profiles (from the Raman lidar observations),

then you should at least take the dust layer optical depth from the Raman lidar observations to constrain the Klett solution. The Klett column backscatter times the used input lidar ratio must match the Raman solution for the dust optical depth. By playing around with the Klett solutions to find the best lidar ratio, you finally end up with the most appropriate column dust layer lidar ratio.

After optimizing the Klett/Raman solution set you may continue with radiation calculations and show remaining differences in terms of TOA and SFC forcings. I am sure they are small.

---

## Referee Comment (RC2) · Anonymous Referee #2 · 20 Aug 2017

First of all, I apologize for my late review.

This manuscript aims to understand and quantify the relative importance between the impact of retrieval method and data processing on radiation at the top of the atmosphere (TOA) and at the surface, used as a measurement/processing guideline for ground-based lidar networks and current/future satellite missions. Analyses and conclusions are based on two case studies, one dust and the other cirrus cloud. While this

manuscript fits the scope of AMT, I feel that more substance is needed for a publication.

1) I am afraid that my main concern is the substance of the manuscript. I strongly support the idea of using radiation as an ultimate evaluation metric, but I feel that the manuscript was submitted too early and that the content is very much on the thin side. To make this manuscript useful, it would be good to address the following issues:

a. The representativeness of cases: I agree it is not necessary to present overwhelming cases, but a synthesis from many cases is needed. This issue becomes even more crucial when the manuscript claims to be "in view of next and current lidar space mission", which is about a global scale and a longer time scale. I like grand statements like that to tell readers what the paper is about, but we also need to be careful not to oversell it. To be scientifically rigorous, I would think that the authors need to get the climatology of dust layer and cirrus clouds (either doing analyses on their own or taking information from the literature) to provide context of whether these two cases represent the majority of the observations, or they are actually outliers. Without that context, we really cannot say much from two cases. Once the climatology is available, then the authors can carefully select cases and think about a strategy how to best cover a wide range of dust/cirrus characteristics.

b. The methodology: The authors recognize the need of actual radiation measurements for their work, but unfortunately, they didn't go further to do it. For ice clouds, there is a BAMS paper http://journals.ametsoc.org/doi/pdf/10.1175/BAMS-88-2-191 talking about radiation closure. Although that paper focused on inter-comparison of various retrieval methods and had a different purpose from the manuscript, it shows how sensitive shortwave/longwave fluxes and radiances are to ice cloud properties. Without comparing with radiation measurements, it is hard to know if the retrieval shown in the manuscript is good enough to be used to provide any recommendation. Additionally, the current form very much just reports numbers of "net radiative forcing" without any discussions. Note that there can be compensating errors from input variables in radiation calculations, so the resulting radiative effects should be discussed in more

detail.

2) The manuscript title is unnecessarily complicated and does not capture the key points. Essentially, this manuscript uses various input aerosol/cirrus properties (from retrieval) to compute radiative fluxes at TOA and at the surface, and then uses these fluxes to evaluate whether retrieval itself or the vertical resolution of profiles plays a more important role in the resulting fluxes. With this objective, radiative effect is the important component, not the choice of the radiative transfer code. Any decent radiative transfer code can do the work. I also don't think proxy is the right word to use. A title is supposed to be precise and to grab attention. For the sake of the authors, I will strongly recommend changing the current title to a simple yet effective one that truly reflects what has been discussed.

3). Following the comment above, it will be better to highlight why the Fu-Liou-Gu code works well for this study. My guess is that it has a rather sophisticated way to characterize optical properties for both aerosol and ice clouds, which is worth mentioning.

4). The misuse of radiative forcing. While some people loosely use radiative forcing and radiative effects and treat them like they are the same, they are, by definition, not the same. I believe what the authors did in the manuscript is calculating radiative effects, not forcing, although no description is ever given in the manuscript. Please clarify and describe it clearly.

5). Referencing could be better. For example, the first paragraph in Introduction should use some proper, more specific citations. And, Page 2, Line 4: Surely, Holben et al. (1998) is the standard citation for AERONET. But to demonstrate "Cloud and aerosol optical properties have been studied... ", papers using AERONET for studying cloud and aerosol should be added here. Also, it would be better to recognize and include studies using ARM or Cloudnet or ACTRiS observations. Same comments for satellite observations.

---

## Referee Comment (RC3) · Anonymous Referee #3 · 23 Aug 2017

General comments

The main objective of this paper, entitled "Fu-Liou Gu radiative transfer model used as proxy to evaluate he impact of data processing and different lidar measurement techniques in view of next and current lidar space missions" is to quantify inconsistences in aerosol (one case in this study : dense dust aerosol event) and cloud (one case in this study : thin cirrus) radiative forcing at Top Of the Atmosphere and at surface due to

two different ground lidar techniques (elastic and Raman lidar, i.e. the Multi-wavelengh System for Aerosols (MUSA) Lidar (Madonna et al., 2011)) and/or data processing (i.e. effect lidar measurement with different vertical resolution together with smoothing techniques). Vertical profiles of aerosols and cloud optical properties (i.e. extinction) are retrieved with classical algorithm (lidar ratio is set to 45 Sr for the aerosol event and to 25 Sr for the cirrus) and with the more accurate Raman lidar techniques. Then radiative forcing is computed with the help of Fu-Liou Gu radiative transfer. Sensitivity of radiative forcing to input parameters (extinction) is evaluated applying a Monte Carlo technique. Aerosol type is the number 17 in the radiative transfer model, and effective diameter of cirrus crystals is computed from Heymsfield et al. (2014) parametrisation. Finally, on the basis of this two study cases, authors conclude that radiative forcing is affected by the measurement and retrieval techniques as well as on the data processing constrainst/assumptions from 0.5% to 35%.

This paper address relevant scientific topics within the scope of AMT. Scientific methodologies and assumptions are valid but not always clearly outlined (see my specific comments). Description of experiments and calculations are rather complete. The overall presentation is rather structured and clear.

Nevertheless, I have two problems when I review this paper. Firstly, even if scientific methodology and calculation are interesting, scientific contribution of this work is not very novel. This paper is rather a sensitivity study of radiative forcing to vertical profiles of extinction retrieved by two different lidar techniques (classic and Raman lidar) but for only two specific two cases (an aerosol event and a thin cirrus). It is also obvious that vertical resolution of lidar measurement (and smoothing techniques) affects computed radiative forcing. I don't understand why these only two cases are representative of the numerous atmospheric conditions. I have the feeling that this paper presents early results and do not reach the scientific level of AMT. Maybe authors could go further in their investigations by, for example, analysing typical atmospheric conditions and/or more extreme atmospheric conditions (cirrus with large optical depth, with different

effective radius, altitude, different aerosols, etc. . .).

Secondly, there is no coherency between the work and results presented in this paper and the title that do not reflect the contents of this paper. Fist off all, the title talk about "next and current lidar space missions". When I read this title and introduction, I expected that authors investigate also the sensitivity of radiative forcing due to the difficulty (spatial and temporal averaging scale) of retrievals of extinctions with CALIOP/CALIPSO or with CATS or with EarthCARE. However, authors refer this fact in the introduction but not in their computations and analyses. Moreover, EarthCARE lidar is a high spectral resolution lidar, witch is not exactly the same technique as the Raman technique. Next, I do not understand why authors make emphasis on the Fu-Liou Gu radiative transfer model. Certainly, this model is a good model. But why this model is considered by the authors as a proxy? Why it is stressed in the title like that?

Finally, number and quality of references are not always appropriate and authors give their own references too often. To conclude, this paper can be accepted with major revisions (see also my specific comments further).

Specific comments

Page 1, line 17 (and further in the text) : Please give the mathematic definition of the net radiative forcing. In general we talk about radiative forcing defined as the change in the net (down minus up) irradiance.

Page2, line 2-3 : references are not appropriate.

Page2, line 3 : Cloud and aerosols have been also studied with POLDER/PARASOL.

Page2, line 21 : Please give other references on the retrievals of aerosol and cloud properties with Raman lidar. By the way, what are the effects of multiple scattering with Raman lidar ? References ?

Page2, line 26, eq 1 : This equation is not well written (exp)

[Figure]

Page 3, line 2 : Please give other references.

Page 4, line3 : Reference of Campbell et al., 2016 is not provided.

Page 4, line 7 : You talk about CATS and EarthCARE. What about the high spectral resolution technique compared to Raman technique?

Page 4 , line 18 : Heymsfield et al. (2014) is not appropriate.

Page 4, line 24 : Why aerosol type number 17. What are optical properties of this aerosol?

Page 5, line 3 : MUSA seem a great lidar, with polarization measurement. Why do not use polarization information in this study ?

Page 5, line 2 : What is the crystal shape of the cirrus ? What is the effect of changing effective diameter on the computed radiative forcing ?

Page 6, line 8 : This cirrus is very optically thin. What is the vertically optical depth ? Why do you choose such a small optical thickness? What is append if optical depth is large (1.5 to 3) ? What about the effect of multiple scattering? Do the retrieval algorithms (classic and Raman) take account of multiple scattering? For space mission lidar data, multiple scattering effects can be not negligible.

---

## Referee Comment (RC4) · Anonymous Referee #4 · 7 Sep 2017

This work deals with the use of different lidar techniques and configurations for studying radiative forcing of aerosol and clouds. In particular, authors analyze the use of backscatter and Raman lidar signals. Backscattering lidar needs the assumption of a constant extinction-to-backscatter lidar ratio for the entire profile while combination of backscattering and Raman signals allow indpendent retrievals of aerosol and clouds extinction and backscattering profiles. Authors show that different lidar techniques and different data processing produce different resuts, and in this research advance

in showing quantitatively how much are those discrepancies. The novelty of this work is then in quantifying the impact of each technique on radiative forcing calculations at TOA and SFC. Due to the large number of backscattering lidar, e.g. MPLNET network uses such systems and very few EARLINET instruments do have Raman lidar during daytime, the results of this analysis are of great interest for the scientific community and valuable for its publication in Atmospheric Measurement Techniques. Neverthleless, I agree with other reviewers that major revisions are needed as the publication suffers from hasty writing and more cases should be considered. Other concerns should be addressed before publication:

MAJOR REVISIONS

1.- I think that a single case thin cirrus cloud is not exhaustive for the analysis. I would rather extend the research at least for three cases: thin cirrus clouds (as already studied) with COD<0.03, Opaque cirrus clouds,with a COD in between 0.03 and 0.3 and thick cirrus cloud case, with a COD>0.3

2.- It comes from the analyis that there is a different behaviour between cirrus cloud and aerosols (cf. fig. 3 and fig. 4) It could be very intresting to add in the analysis cases where there is a simultaneous presence of a cirrus cloud on top of an aerosol layer, like dust or biomass-burning. In those cases it would be interesting to verify if technique or data processing are critical

MINOR REVISIONS

3.- The description of lidar signals and the different ways of resolving the equations should be in a methodology section.

4.- Page 2, line 22: Traditional lidar Raman are expensive but the development of the rotational Raman techniques make it cheaper and improve signal-to-noise. Please include it in your discussion.

5.- Page 2, line 23: The High Spectral Resolution Lidar and Dial techniques should be

commented and cited.

6.- The NASA Aerosol-Clouds-Ecosystems mission does plan to implement a multiwavelength HSRL system in the space allowing retrievals of aerosol microphysical parameters. Please include it in your discussion.

7.- Radiative transfer codes do assume certain aerosol properties for each species. The Fu-Liou-Gu model assumes OPAC aerosol module, which may differ from real measurements. Retrievals of aerosol microphysical properties can improve retrievals of radiative forcing if aerosol effective radius and single scattering albedo are introduced. Please discuss the use of an aerosol model

8.- I agree with the previous referees that the current title does not macth approately with the goal of the manuscript. Please consider to change it.

---

## Author Comment (AC1) · 12 Nov 2017

**Statement on the Revision of ⟨AMT 2017-182⟩ Based on the Referees' Report**

S. Lolli     F. Madonna     M. Rosoldi     J. R. Campbell,
E. J Welton     J.R. Lewis     Y. Gu     G. Pappalardo

November 12, 2017

This statement concerns our revision of the ⟨AMT 2017-182⟩ paper, entitled "⟨*Fu-Liou Gu radiative transfer model used as …*⟩", based on the referees' report.

**Comments by Reviewer #1**

> Although the paper deals with lidar observations of cirrus
> extinction profiles, there is no information on the laser
> beam pointing (zenith or off zenith to avoid specular
> reflection) and no information about the receiver field
> of view which has an impact on the multiple scattering
> contribution.  On the other side, the depolarization
> technique is explained (even the 45 deg calibration)
> although not used.  Please re-write this section, update
> the instrument part to meet the requirements for this paper.

The information about instrument depolarization channel was suppressed as not relevant for the paper, being the channel not used. On the contrary, we added a paragraph regarding measurement configuration and multiple scattering effects.

> ```
> Now, I come to my most important point:  The authors use
> both, the Raman lidar method and the Klett retrieval to
> determine particle extinction profiles.  And EARLINET
> members (experts in the field of Raman lidars) probably
> know that the optimum Klett solutions of the backscatter
> and the extinction profiles are obtained with the 'actual'
> lidar ratio (profile) from the Raman lidar observations.
> Ideally, Klett and Raman backscatter and extinction profiles
> coincide, ...  but usually the available Klett codes cannot
> handle lidar ratio profiles.  However, if you apply the
> method to such a rather rather thin cirrus as done in this
> paper, then we may have a problem.  I would recommend to use
> a visible, very well developed cirrus cloud deck (not this
> subvisible cirrus with an optical depth 0f about 0.02).  Is
> there a reason why this quite unusual cirrus is taken, and
> not a very normal one?
> ```

In the manuscript new version two more cases are reported and discussed, a thicker cirrus cloud and a case with biomass burning aerosol. Regarding the first part of the comment, we changed the text accordingly to make clear that the goal of this study it is to start a relevant discussion from a quantitative point of view, about the discrepancies of aerosol and cloud direct radiative effect calculated using the Raman technique or the simpler lidar elastic technique retrievals. Inconsistencies may arise also using a mixture of lidar techniques from multiple networks or within the same network. As example, what is the difference in retrieval if, we have data from an MPLNET permanent observation station vs. a more sophisticated (like those operating in the frame of EARLINET) instrument? This first work put the basis for a successive study where a much larger dataset will be analyzed to assess quantitatively how much the different techniques/data processing affect the retrieval of the optical and geometrical properties.

> Nevertheless, by just taking a climatological value for the
> dust lidar ratio of 45 sr and for the cirrus of 25 sr in
> the Klett retrievals, and in this way by completely ignoring
> the reality, i.e., the 'actual' Raman lidar observations
> of the lidar ratio .  .  .. it is not surprizing that you
> obtain different Klett and Raman extinction profiles.  The
> true ones are, by the way, the Raman solutions.  The Klett
> solutions are wrong.  If your Klett code cannot handle lidar
> ratio profiles (from the Raman lidar observations), then you
> should at least take the dust layer optical depth from the
> Raman lidar observations to constrain the Klett solution.
> The Klett column backscatter times the used input lidar
> ratio must match the Raman solution for the dust optical
> depth.  By playing around with the Klett solutions to find
> the best lidar ratio, you finally end up with the most
> appropriate column dust layer lidar ratio.  After optimizing
> the Klett/Raman solution set you may continue with radiation
> calculations and show remaining differences in terms of TOA
> and SFC forcings.  I am sure they are small.

Thanks for pointing it out but again, we think that we didn't state clearly enough the scope of our manuscript. We revised the text to avoid any possible confusion or misunderstanding. This study is preparatory for a future standardization of existing or future ground-based lidar network using different techniques as well space missions. The used metric for this evaluation is the net radiative effect calculation at TOA and SFC by the Fu-Liou-Gu radiative transfer model. The manuscript focuses on discrepancies between lidar techniques/data processing, not on the assumptions of the single retrieval of aerosol/cloud geometrical optical properties. Theoretically, the analysis can be performed on synthetic signals where all the geometrical, optical and microphysical cloud and aerosol properties are well known. In future work, a quantitative assessment of the differences will be evaluated on real cases taken from a climatological significant database

---

## Author Comment (AC2) · 12 Nov 2017

**Comments by Reviewer #2**

1) I am afraid that my main concern is the substance of the
manuscript. I strongly support the idea of using radiation
as an ultimate evaluation metric, but I feel that the
manuscript was submitted too early and that the content is
very much on the thin side. To make this manuscript useful,
it would be good to address the following issues: a. The
representativeness of cases: I agree it is not necessary
to present overwhelming cases, but a synthesis from many
cases is needed. This issue becomes even more crucial when
the manuscript claims to be \in view of next and current
lidar space mission", which is about a global scale and a
longer time scale. I like grand statements like that to
tell readers what the paper is about, but we also need to be
careful not to oversell it. To be scientifically rigorous,
I would think that the authors need to get the climatology
of dust layer and cirrus clouds (either doing analyses on
their own or taking information from the literature) to
provide context of whether these two cases represent the
majority of the observations, or they are actually outliers.
Without that context, we really cannot say much from two
cases. Once the climatology is available, then the authors
can carefully select cases and think about a strategy how to
best cover a wide range of dust/cirrus characteristics.

We would like to thank the reviewer for the meaningful comment. However, if from one side we agree that two cases are not enough (for this reason we added two more cases with an opaque cirrus clouds and a biomass burning event) on the other, we we were not trying to oversell our research but think that our manuscript lacks of clarity because the main goal is to evaluate the differences in term of net radiative effects among that the more sophisticated and simpler different lidar techniques. In theory, for the purpose of this manuscript, it can be used synthetic signals instead of real measurements, where the optical and geometrical aerosol and cloud properties are well known and quantify how the lidar technique/data processing affects the raidative transfer calculation, using FLG as metric. Our cases aims to show the existence of these not negligible differences arising from the diversity of lidar techniques/data processing, for the first time quantitatively. The statement that the reviewer is happy with the use of an RTM as the metric to assess the systematic effects in the retrieval of aerosol forcing using lidar is a strong encouragement for us to continue this work and assess the impact on much larger dataset.

```
b.  The methodology:  The authors recognize the
need of actual radiation measurements for their
work, but unfortunately, they didn't go further
to do it.  For ice clouds, there is a BAMS paper
http://journals.ametsoc.org/doi/pdf/10.1175/BAMS-88-2-191
talking about radiation closure.  Although that paper
focused on intercomparison of various retrieval methods
and had a different purpose from the manuscript, it shows
how sensitive shortwave/longwave fluxes and radiances are
to ice cloud properties.  Without comparing with radiation
measurements, it is hard to know if the retrieval shown
in the manuscript is good enough to be used to provide any
recommendation.  Additionally, the current form very much
just reports numbers of \net radiative forcing" without any
discussions.  Note that there can be compensating errors
from input variables in radiation calculations, so the
resulting radiative effects should be discussed in more
detail.
```

We agree that the microphysics parameterization of the cirrus cloud plays a fundamental role in calculating the net radiative effects of cirrus clouds and aerosol layers. For our calculations we used the empiric parameterization as found in Heymsfield et al., 2014. As stated in the paper mentioned in the comment, each parameterization shows pros and cons. However, as stated in the previous answer, our analysis can be carried out in principle on synthetic signal where the microphysics is fully known and still quantitatively describe the differences for the different retrievals. In fact, we are interested in the relative values between different lidar techniques/data processing. To reach this goal we use a RTM on the different retrieval and calculate the relative discrepancies (we applied the same parameterization for all the retrieved profiles). In future analysis we are going to take into consideration different parameterizations. Nevertheless we added some additional paragraphs where we clarify our choice and state how different parameterizations can affect the results citing properly the suggested BAMS paper.

> 2). The manuscript title is unnecessarily complicated
> and does not capture the key points. Essentially, this
> manuscript uses various input aerosol/cirrus properties
> (from retrieval) to compute radiative fluxes at TOA and at
> the surface, and then uses these fluxes to evaluate whether
> retrieval itself or the vertical resolution of profiles
> plays a more important role in the resulting fluxes. With
> this objective, radiative effect is the important component,
> not the choice of the radiative transfer code. Any decent
> radiative transfer code can do the work. I also don't think
> proxy is the right word to use. A title is supposed to be
> precise and to grab attention. For the sake of the authors,
> I will strongly recommend changing the current title to a
> simple yet effective one that truly reflects what has been
> discussed.

Agreed that the word "proxy" is misused and generates confusion. For this reason we changed completely the manuscript title into a simpler form:"Impact of the different lidar measurement techniques and data processing on evaluating cirrus cloud and aerosol direct radiative effects.". This new title version we think is simple and clear and really reflects what has been done in the paper.

> 3). Following the comment above, it will be better to
> highlight why the Fu-Liou-Gu code works well for this
> study. My guess is that it has a rather sophisticated way
> to characterize optical properties for both aerosol and ice
> clouds, which is worth mentioning.

Agreed, we added a paragraph to describe in detail how the Fu-Liou-Gu radiative transfer model works and why it works good for reach the objective stated in the manuscript.

> 4) The misuse of radiative forcing. While some people
> loosely use radiative forcing and radiative effects and
> treat them like they are the same, they are, by definition,
> not the same. I believe what the authors did in the
> manuscript is calculating radiative effects, not forcing,
> although no description is ever given in the manuscript.
> Please clarify and describe it clearly.

We agree that the word "forcing" is often misused. Of course we calculate the net radiative effect of cirrus clouds and aerosol layers. We added a paragraph to describe the computation we performed and we substitute in the entire manuscript the word "forcing" with "effect".

6

> 5). Referencing could be better. For example, the first
> paragraph in Introduction should use some proper, more
> specific citations. And, Page 2, Line 4: Surely, Holben
> et al. (1998) is the standard citation for AERONET. But
> to demonstrate \Cloud and aerosol optical properties have
> been studied. . . \, papers using AERONET for studying
> cloud and aerosol should be added here. Also, it would
> be better to recognize and include studies using ARM
> or Cloudnet or ACTRiS observations. Same comments for
> satellite observations.

We agree and we changed accordingly the manuscript adding and acknowledging ARM, Cloudnet and ACTRIS work.

---

## Author Comment (AC3) · 12 Nov 2017

**Comments by Reviewer #3**

General comments The main objective of this paper, entitled
\Fu-Liou Gu radiative transfer model used as proxy to
evaluate he impact of data processing and different lidar
measurement techniques in view of next and current lidar
space missions" is to quantify inconsistences in aerosol
(one case in this study :  dense dust aerosol event) and
cloud (one case in this study :  thin cirrus) radiative
forcing at Top Of the Atmosphere and at surface due to
two different ground lidar techniques (elastic and Raman
lidar, i.e.  the Multi-wavelengh System for Aerosols (MUSA)
Lidar (Madonna et al., 2011)) and/or data processing (i.e.
effect lidar measurement with different vertical resolution
together with smoothing techniques).  Vertical profiles of
aerosols and cloud optical properties (i.e.  extinction)
are retrieved with classical algorithm (lidar ratio is set
to 45 Sr for the aerosol event and to 25 Sr for the cirrus)
and with the more accurate Raman lidar techniques.  Then
radiative forcing is computed with the help of Fu-Liou
Gu radiative transfer.  Sensitivity of radiative forcing
to input parameters (extinction) is evaluated applying a
Monte Carlo technique.  Aerosol type is the number 17 in
the radiative transfer model, and effective diameter of
cirrus crystals is computed from Heymsfield et al.  (2014)
parametrisation.  Finally, on the basis of this two study
cases, authors conclude that radiative forcing is affected
by the measurement and retrieval techniques as well as
on the data processing constrainst/assumptions from 0.5%
percent to 35% This paper address relevant scientific
topics within the scope of AMT. Scientific methodologies
and assumptions are valid but not always clearly outlined
(see my specific comments).  Description of experiments and
calculations are rather complete.  The overall presentation
is rather structured and clear.

We thank the reviewer for the positive comments

> Nevertheless, I have two problems when I review this paper.
> Firstly, even if scientific methodology and calculation
> are interesting, scientific contribution of this work is
> not very novel.  This paper is rather a sensitivity study
> of radiative forcing to vertical profiles of extinction
> retrieved by two different lidar techniques (classic
> and Raman lidar) but for only two specific two cases (an
> aerosol event and a thin cirrus).  It is also obvious that
> vertical resolution of lidar measurement (and smoothing
> techniques) affects computed radiative forcing.  I don't
> understand why these only two cases are representative of
> the numerous atmospheric conditions.  I have the feeling
> that this paper presents early results and do not reach the
> scientific level of AMT. Maybe authors could go further in
> their investigations by, for example, analysing typical
> atmospheric conditions and/or more extreme atmospheric
> conditions (cirrus with large optical depth, with different
> effective radius, altitude, different aerosols, etc.  .  .).

We agree that it is already known that different lidar techniques and data processing produce different results, but in literature a discussion on the uncertainty/impact due to the use of different lidar techniques to validate the radative forcing inferred from satellite platform or modeling measurements is indeed missing. As metric we used the Fu-Liou-Gu radiative transfer model net radiative effect at the Top of the Atmosphere (for satellite based measurements) and at surface (for ground based measurements). Even if in literature many studies are based on case studies, we agree that the presented case are not enough. For this reason we added two more cases: one including a biomass burning event and another a thick cirrus cloud.

> Secondly, there is no coherency between the work and results
> presented in this paper and the title that do not reflect
> the contents of this paper. Fist off all, the title talk
> about \next and current lidar space missions". When I
> read this title and introduction, I expected that authors
> investigate also the sensitivity of radiative forcing due
> to the difficulty (spatial and temporal averaging scale)
> of retrievals of extinctions with CALIOP/CALIPSO or with
> CATS or with EarthCARE. However, authors refer this fact
> in the introduction but not in their computations and
> analyses. Moreover, EarthCARE lidar is a high spectral
> resolution lidar, witch is not exactly the same technique as
> the Raman technique. Next, I do not understand why authors
> make emphasis on the Fu- Liou Gu radiative transfer model.
> Certainly, this model is a good model. But why this model
> is considered by the authors as a proxy? Why it is stressed
> in the title like that?

We agree that the title can generate confusion and the manuscript lacks of clarity in this sense. For this reason we specified it in the title and changed the text accordingly. The rationale behind the title is that we would like to raise awareness on how much the different lidar techniques/data processing affect the retrieval of the optical and geometrical properties of the aerosol and cloud layers, bearing in mind that also several space missions are going on and other are ready to be launched using these techniques/data processing. We changed completely the title into:"Impact of the different lidar measurement techniques and data processing on evaluating cirrus cloud and aerosol direct radiative effects."

> Specific comments Page 1, line 17 (and further in the
> text) : Please give the mathematic definition of the net
> radiative forcing. In general we talk about radiative
> forcing defined as the change in the net (down minus up)
> irradiance.

We provided in the text the definition of direct radiative effect accordingly. For this study we used the difference between the total sky (when cloud and/or aerosols are present) and the pristine sky (clear atmosphere)

> Page2, line 2-3 : references are not appropriate.

The provided references investigate how the sign in net radiative effect of cirrus clouds can change daytime. Then, the net forcing is still uncertain.

> Page2, line 3 :  Cloud and aerosols have been also studied
> with POLDER/PARASOL.

References were added

> Page2, line 21 :  Please give other references on the
> retrievals of aerosol and cloud properties with Raman lidar.
> By the way, what are the effects of multiple scattering with
> Raman lidar ?  References ?

References were added. Multiple scattering is of course playing an important role mostly for clouds. However, investigating multiple scattering is beyond the scope of the manuscript as we start our analysis using the available products. As the answer given for another reviewer, we try to quantify only the technique/data processing discrepancy, not other effects. For the purpose of the manuscript, also synthetic signals can be used.

> Page2, line 26, eq 1 :  This equation is not well written
> (exp)

Changed accordingly

> Page 3, line 2 :  Please give other references.

Additional references are provided

> Page 4, line3 :  Reference of Campbell et al., 2016 is not
> provided.

The reference is now provided

> Page 4, line 7 :  You talk about CATS and EarthCARE. What
> about the high spectral resolution technique compared to
> Raman technique?

That's an interesting point. Unfortunately, in this first study we don't have co-located HSRL measurements to compare.

> Page 4 , line 18 :  Heymsfield et al.  (2014) is not
> appropriate.

Fixed

> Page 4, line 24 : Why aerosol type number 17. What are
> optical properties of this aerosol?

This type of aerosol is labeled as transported dust. However, we are interested in relative discrepancies, as we use for all the cases this aerosol type. We agree that the absolute value may be incorrect.

> Page 5, line 3 : MUSA seem a great lidar, with polarization
> measurement. Why do not use polarization information in
> this study ?

Actually all the information obtained from MUSA lidar observations, i.e. the geometrical and optical properties of aerosols and clouds at different wavelengths together with depolarization and ancillary information (e. g. back-trajectories) were used to identify aerosol type and cloud phase. While only the aerosol/cloud extinction profile is used as input for the FLG radiative transfer model.

> Page 5, line 2 : What is the crystal shape of the cirrus
> ? What is the effect of changing effective diameter on the
> computed radiative forcing ?

We use Heymsfield et al., 2014 empirical parameterization. Again, as we are interested in relative values of the net radiative effect, the parameterization is not fundamental for our analysis because it is the same for the considered lidar techniques/data processing.

> Page 6, line 8 : This cirrus is very optically thin. What
> is the vertically optical depth ? Why do you choose such
> a small optical thickness? What is append if optical depth
> is large (1.5 to 3) ? What about the effect of multiple
> scattering? Do the retrieval algorithms (classic and
> Raman) take account of multiple scattering? For space
> mission lidar data, multiple scattering effects can be not
> negligible.

We added a case with an optically thicker cirrus cloud. For sure, the multiple scattering affects mainly the cirrus cloud net radiative effect calculations, as the multiple scattering is modifying the cloud atmospheric extinction profile. However, in this first study, the different techniques and data processing profiles are not corrected by multiple scattering effects, as we are interested in quantifying

the relative differences. For the scope it can be used a synthetic cloud signal where multiple scattering effects are not present.

**Comments by Reviewer #4**

> This work deals with the use of different lidar techniques
> and configurations for studying radiative forcing of
> aerosol and clouds.  In particular, authors analyze the
> use of backscatter and Raman lidar signals.Backscattering
> lidar needs the assumption of a constant extinction-to
> backscatter lidar ratio for the entire profile while
> combination of backscattering and Raman signals allow
> independent retrievals of aerosol and clouds extinction
> and backscattering profiles.  Authors show that different
> lidar techniques and different data processing produce
> different results, and in this research advance in showing
> quantitatively how much are those discrepancies.  The
> novelty of this work is then in quantifying the impact
> of each technique on radiative forcing calculations at
> TOA and SFC. Due to the large number of backscattering
> lidar, e.g.  MPLNET network uses such systems and very few
> EARLINET instruments do have Raman lidar during daytime,
> the results of this analysis are of great interest for the
> scientific community and valuable for its publication in
> Atmospheric Measurement Techniques.  Nevertheless, I agree
> with other reviewers that major revisions are needed as the
> publication suffers from hasty writing and more cases should
> be considered.  Other concerns should be addressed before
> publication:  1.- I think that a single case thin cirrus
> cloud is not exhaustive for the analysis.  I would rather
> extend the research at least for three cases:  thin cirrus
> clouds (as already studied) with COD<0.03, Opaque cirrus
> clouds,with a COD in between 0.03 and 0.3 and thick cirrus
> cloud case, with a COD>0.3

Thanks for the meaningful comment. We added a thicker cirrus cloud in the analysis.

---

## Author Comment (AC4) · 12 Nov 2017

the relative differences. For the scope it can be used a synthetic cloud signal where multiple scattering effects are not present.

**Comments by Reviewer #4**

> This work deals with the use of different lidar techniques
> and configurations for studying radiative forcing of
> aerosol and clouds.  In particular, authors analyze the
> use of backscatter and Raman lidar signals.Backscattering
> lidar needs the assumption of a constant extinction-to
> backscatter lidar ratio for the entire profile while
> combination of backscattering and Raman signals allow
> independent retrievals of aerosol and clouds extinction
> and backscattering profiles.  Authors show that different
> lidar techniques and different data processing produce
> different results, and in this research advance in showing
> quantitatively how much are those discrepancies.  The
> novelty of this work is then in quantifying the impact
> of each technique on radiative forcing calculations at
> TOA and SFC. Due to the large number of backscattering
> lidar, e.g.  MPLNET network uses such systems and very few
> EARLINET instruments do have Raman lidar during daytime,
> the results of this analysis are of great interest for the
> scientific community and valuable for its publication in
> Atmospheric Measurement Techniques.  Nevertheless, I agree
> with other reviewers that major revisions are needed as the
> publication suffers from hasty writing and more cases should
> be considered.  Other concerns should be addressed before
> publication:  1.- I think that a single case thin cirrus
> cloud is not exhaustive for the analysis.  I would rather
> extend the research at least for three cases:  thin cirrus
> clouds (as already studied) with COD<0.03, Opaque cirrus
> clouds,with a COD in between 0.03 and 0.3 and thick cirrus
> cloud case, with a COD>0.3

Thanks for the meaningful comment. We added a thicker cirrus cloud in the analysis and a biomass burning aerosol event.

> It comes from the analyis that there is a different behavior
> between cirrus cloud and aerosols (cf. fig. 3 and fig. 4)
> It could be very intresting to add in the analysis cases
> where there is a simultaneous presence of a cirrus cloud on
> top of an aerosol layer, like dust or biomass-burning. In
> those cases it would be interesting to verify if technique
> or data processing are critical

We agree with the reviewer that a simultaneous presence of clouds and aerosol layers could be very interesting, but in our analysis is limited to single layer analysis to avoid any error compensation due to multiple mode.

> 3.- The description of lidar signals and the different
> ways of resolving the equations should be in a methodology
> section.

We added it accordingly

> 4.- Page 2, line 22: Traditional lidar Raman are expensive
> but the development of the rotational Raman techniques make
> it cheaper and improve signal-to-noise. Please include it
> in your discussion.

Even if we didn't go further in the analysis, we added a paragraph describing rotational Raman lidar adding also a reference: Veselovskii, I., Whiteman, D.N., Korenskiy, M., Suvorina, A., Pérez-Ramírez, D., (2015) Use of rotational Raman measurements in multiwavelength aerosol lidar for evaluation of particle backscattering and extinction. Atmospheric Measurement Techniques 8, 4111-4122.

> 5.- Page 2, line 23: The High Spectral Resolution Lidar and
> Dial techniques should be commented and cited.

Added accordingly to the text.

> 6.- The NASA Aerosol-Clouds-Ecosystems mission does plan
> to implement a multiwavelength HSRL system in the space
> allowing retrievals of aerosol microphysical parameters.
> Please include it in your discussion.

A short paragraph was added describing ACE and referenced (Whiteman, D.N., Pérez-Ramírez, D., Veselovskii, I., Colarco, P., Buchard, V. (2017) Simulations

of spaceborne multiwavelength lidar measurements and retrievals of aerosol microphysics. Journal of Quantitative Spectroscopy and Radiative Transfer, submitted.)

> 7.- Radiative transfer codes do assume certain aerosol
> properties for each specie.  The Fu-Liou-Gu model
> assumes OPAC aerosol module, which may differ from
> real measurements.  Retrievals of aerosol microphysical
> properties can improve retrievals of radiative forcing if
> aerosol effective radius and single scattering albedo are
> introduced.  Please discuss the use of an aerosol model

We agree with the referee. However, retrievals of aerosol microphysical properties require multi-wavelength lidar (e.g. Veselovskii et al., 2002, 2015), which are very sophisticated instrument sensitive to systematic and random errors in the optical data (Perez-Ramirez et al., 2013). Because we focus on lidar systems that can operate continuously in different networks, and our radiative forcing calculations do not vary much when changing effective radius and single scattering albedo.

> 8.- I agree with the previous referees that the current
> title does not macth approately with the goal of the
> manuscript.  Please consider to change it.

Changed accordingly.

---

## Referee Report (RR1)

The authors have addressed most of the concerns I had before recommending the article for its publication in Atmospheric Measurement Techniques. Personally, I think that the new title proposed is appropriate and that the new study cases for aerosol and cirrus clouds show more convincing results. But I have some minor concern I would like authors address before recommending the final publication:

- Although the authors responded well to my concern about the effects of aerosol microphysical properties in aersol radiative forcing computations, I miss such a paragraph in the revised manuscript.
- Authors refer in the abstract and in the text to aerosol and clouds optical and geometrical properties. Please, replace by 'aerosol and clouds optical and microphysical properties'.
- Although HSRL technique is not used in operational lidar networks such as EARLINET, their potential can not be ignored. Please refer this in the text and add appropriate references.
- The statement about the upcoming NASA Aerosol-Clouds-Ecosystems mission is not in the correct place. It is very important to reference such mission, and I recommend to move the sentence at the end of line 141. Also, please update reference to Whiteman et al., 2018
    - Whiteman, D.N., Pérez-Ramírez, D., Veselovskii, I., Colarco, P., Buchard, V. (2018) Simulations of spaceborne multiwavelength lidar measurements and retrievals of aerosol microphysics. Journal of Quantitative Spectroscopy and Radiative Transfer, 205, 27-39
- Please, define each term of equation 4.
- Axis of Figure 1 and Figure 2 are difficult to read.

---

## Author Response (AR2)

**Statement on the Revision of (AMT 2017-182) Based on the Referees' Report**

S. Lolli F. Madonna M. Rosoldi J. R. Campbell, E. J Welton J.R. Lewis Y. Gu G. Pappalardo

February 15, 2018

This statement concerns our revision of the  $\langle AMT 2017-182 \rangle$  paper, entitled " $\langle Impact of Varying Lidar Measurement and Data Processing Techniques... \rangle$ ", based on the referees' report.

**Comments by Reviewer #1**

The revised version is not ready for publication. The quality is rather low, and if this would be the first round of review, I would vote for rejection. I am a bit disappointed about the answers of the first author (I speculate the co-authors were never involved in the review). I have the feeling after reading the responses, the first author is not willing to invest more work, needed to become a good science article. This is disappointing.

We would like to thank the reviewer for revising the paper again, and for the helpful suggestions for improving the quality of the paper. In the following specific answers to all the points raised by the reviewer are addressed.

As general remark we want to reassure the reviewer that all the revisions have been carefully discussed among all the co-authors.

Zenith pointing of the laser beam (this is obviously the case for the Italian lidar, rather than the required off-zenith pointing) and multiple scattering can introduce severe uncertainties in cirrus extinction profiling with lidar so that all the subsequent radiative transfer computations are useless when the multiple scattering effect is not considered and, independent of that, very questionable when the lidar is pointing to the zenith. As a consequence, the cirrus studies in this paper are practically useless. More details are given below. The paper is clearly not state-of the-art from the basic and fundamental lidar point of view. At least, major revisions are required.

Those two main issues are now fixed (see discussion below). Again, the previous version was a first study with the goal of showing discrepancies of lidar technique and data processing that can be performed also on synthetic lidar signals.

L27: ... with respect to cirrus... must be added because in the case of liquid-water clouds the single-scattering lidar ratio is 18 sr, and when considering multiple scattering the apparent one is around 8-12 sr.

Added, as suggested by the reviewer.

L45-47: strange argumentation.... to link geometrical properties of aerosols and clouds with passive remote sensing.... which have practically no profiling capability (only the model behind all the column integrated measurements may allow the retrieval of geometrical properties, but then with large error bars).

The sentence has been modified to avoid any possible misunderstanding

L68-70: I would remove these lines and Madonna references. Almost all experts in this field do not believe that the measurements and conclusions are ok.

The authors don't agree with the reviewer. The cited paper went through a regular and independent review process on a top level journal in geosciences (i.e. JGR). The authors believe that it is not fair to open in this context a discussion on the reliability of Madonna's paper results. This is in hand of the scientific community and it is appropriate anyhow to cite it. The authors also would like

to remind to the the reviewer and the editor that this is not the only paper dealing with aerosol observations using radar and that the papers results have been cited by other independent papers in literature also demonstrating the reliability of the Madonna's paper observations.

L90: I do not know why we have equations in the introduction, I would move them to section 2 (2a: lidar methods). And why do we need the simple lidar equation? And why is there the overlap term missing (O(r))? The overlap profile affects MPL observations up to 4-5 km height. That should be mentioned. In such a Section 2a, I would present the Fernald equation and the Raman lidar equation. They are used later on, and in this new Section 2a, all problems and differences could be explained in detail..., which is not possible (or should not be given) in the introduction.

The equation, enriched with overlap term and relative description has been moved to the new section 2.1. MPL overlap is mentioned too.

L99: There are meanwhile so many lidar ratio papers (real measurements!).... Please provide some references, e.g. Muller 2007, Gross, Tellus 2009, ACP 2013, 2015? , Ferrare, JGR..., Sakai, JGR... Veselovskii 2016 and many many others . Ackermann (1998) is just a very simple simulation study.

We take into account the reviewer suggestions and we added the references.

L105: Raman roto-vibrational ... bad wording...

The sentence was rephrased

L133: The impact of these differences on end-user applications have never been evaluated...Please be more precise, what do you mean with end-user applications?

The paragraph was rephrased to avoid ambiguity

L195: Now the RFOV is mentioned. The RFOV is 1.5 mrad. This means the multiple scattering effect is rather large in case of clouds. The measured apparent cirrus extinction coefficient is roughly a factor of 2 lower than the desired single-scattering extinction coefficient (throughout the cirrus layer). The single-scattering extinction coefficient is the basic input in radiative transfer equations. Multiple scattering depends on cirrus crystal microphysics (size, shape, amount...). And such a huge multiple scattering effect is not considered in the paper? ... the authors tell us: they ignore it! This is not acceptable! These effects must be considered. There is no way to circumvent this problem.

We corrected Raman extinction profiles using Eloranta MS code. We used a monodisperse averaged value for the cirrus cloud size diameter obtained from Heymsfield et al., 2014 parameterization. For the two cirrus clouds we obtained a MS corrected value of 24sr and 26sr respectively for 10 June 2010 and 17 Feb 2014. The elastic channel MS correction is restricted to the values of the Lidar Ratio that multiply the retrieved backscattering coefficient (that is barely influenced by MS). In this case, the LR values employed in the analysis are a good estimate from the MS corrected Raman profiles.

L202: I complained last time with good reasons: There is no information on the laser beam pointing (zenith or off zenith). Even in the revised version there is no hint on beam pointing. I speculate the laser beams were pointed to the zenith. This in turn means that the cirrus backscatter profiles are probably strongly affected by specular reflection. In case of zenith pointing, the backscatter coefficients (from which the extinction coefficients are estimated when using the Fernald method) are an order of magnitude larger in the case of aligned, falling crystals than the true backscatter values (obtained at off zenith pointing, 3-5 degrees off the zenith is sufficient). The alignment effect depends on particle size (only >100 mikrometer particles are able to be horizontally oriented), thus the effect can vary from height to height. So, cirrus backscatter and extinction profiles from zenith pointing elastic backscatter lidars are highly corrupt and uncertain, and to my opinion these Fernald extinction profiles are useless for further use in radiative transfer computations.

MUSA lidar is not tilted due to some technical constraints. However, MS corrected Raman channel retrieval show lidar ratio values higher than 20sr.

Those high values support the hypothesis that strong specular reflection is extremely unlikely. In support to this hypothesis, Heymsfield parameterization shows average diameter values ; 100 micrometer. Hogan and Illigworth Moreover, Hogan et Illingworth (2003), found that specular reflection tends to be much stronger and more common when temperatures are between 250 K and 264 K (a temperature range found at much lower altitude with respect to the examined cirrus cloud cases), where plate crystals, which give the greatest specular signal, grow in this temperature range. For this reason we believe that specular reflection doesn't provide a significant contribution to the backscattering profile.

Figure 1: The aerosol measurements are ok. Multiple scattering effects can be neglected, as well as beam pointing effects.Why are there different descriptions (1b, left: Iterative, Iterative, Raman, 1b, right: Raman, Elastic, Elastic) ? X-axis and y-axis text and numbers of Fig 1a are much too small! Figure 2b, left: extinction values of 0.1, 0.2 ...., 0.4 m-1 are wrong... Figure 3: All plot frames should have the same size (and should be well arranged above each other), all x-axis and y-axis text should use the same letter font, and PT The caption should briefly explain what Raman, Full Res., and Iterative means. Full Res is obviously Iterative (60 m), and Iterative denotes Iterative (360 m)? Please harmonize this with the other figures... Figure 4: The same concerning plot frames, and explanations of Raman, Full. Res. and Iterative. X-axis and y-axis text and numbers need to be harmonized. To repeat: The cirrus results in Figure 4 are highly questionable. To my opinion, they are useless because of the lidar-related problems with zenith pointing and multiple scattering.

All plot frames should have exactly the same size (at least for 1a and for 1b)

All the figures have been accordingly modified and corrected, harmonized as the caption and legends.

**Comments by Reviewer #2**

The main objective of this revised paper, novelty entitled \Impact of varying lidar measurement and data processing techniques in evaluating cloud and aerosol direct radiative effects" is still to quantify inconsistencies in aerosol (two case this time in the revised paper : dust aerosol and biomass burning event) and cirrus cloud (two cases in the revised paper : a thin and an opaque cirrus) radiative forcing at Top Of the Atmosphere and at surface due to two different ground lidar techniques (elastic and Raman lidar, i.e. the Multi-wavelengh System for Aerosols (MUSA) Lidar (Madonna et al., 2011)) and/or data processing (i.e. effect lidar measurement with different vertical resolution together with smoothing techniques). The revised paper is based on the same approach and analysis technics as the initial paper, that is a good point. Quality of references of the revised paper are now better and the title is now appropriate. This revised paper still address relevant scientific topics within the scope of AMT. Nevertheless, the revised paper shows dramatic drawback and inconsistency with the study based on cirrus cloud. Figures a not all clear. For a revised paper, it is quite annoying. This revised paper needs major revision. Major remarks: 1) Line 689 (i.e. figure 2) Right panels show the opaque cirrus. On the right top panel, the base and top altitude are between 6 km and 8 km, respectively, whereas the retrieved extinction coefficient profile is between 11 and 13.5 km. Where is the problem?

Thank you for pointing this out. We put by error the old picture. We substitute it and now we have the correct picture for cirrus cloud on 10 June 2010.

2) Line 689((i.e. figure 2) Optical depth is defined as extinction times distance. Bases on this simple formula and on information provided by the figure 2, optical depth tau of thin cirrus, is, roughly, tau=0.1(m-1)\*100(m) = 10 and the optical depth of the opaque cirrus, is, tau=5e-5(m-1)\*2000 (m)= 0.1 ! How can we qualify a thin cirrus with an optical depth of 10 and an opaque cirrus with an optical depth of 0.1 ?

Thanks, nice catch. We changed the figure as there was an error in the measurement unit scale. We added also the new profiles corrected for the multiple scattering. All the pictures are now at high resolution. 3) Line 689((i.e. figure 2) The comparison of the direct radiaitve effect of theses both cirrus is not suitable. Indeed, direct radiative effect is function of optical depth, effective radius (or asymmetric parameter and single scattering albedo) and of temperature (i.e. of the mean altitude of the cirrus but also of the cirrus cloud base and top altitude). In order to interpret differences of direct radiaitve effect between an optically thin and opaque cirrus, geometrical thickness must be the same, as well as the cloud base and cloud top altitude. Moreover, why two different vertical resolution (420 m for the thin cirrus and 780 m for the opaque cirrus? Based on my major comments, it is impossible to evaluate the conclusions of this revised work. This paper need major revision.

The main goal of the manuscript is not to compare the differences in terms of direct radiative effects of the two cirrus clouds, but to assess how much different techniques/data processing affect directly the direct radiadive effects computed on the same cirrus cloud. We performed the analysis on two separate cirrus clouds to show if it is possible to detect any variability with cirrus thickness. We changed the narrative in the manuscript to make it more clear.

Specific comments line 31-32 : This sentence is not clear. Please rephrase line 48-52 : acronyms are missing line 122 : please define lamba 0 line 144 : I think \...computed using..." is better than \... retrieved using..." line 166 : the FLG model needs the phase function ? How is computed the phase function? line 192 : please define a.s.l. Figure 1 : - please define Tito(PZ) - All the \no-information" (above 7 km, the blue color) is useless. Please rescale the figures. - Please use the same color for the legends on right and left (b) - What is the difference between iterative and elastic (on the legend of figure 1b)? - Why the range corrected signal is at 1065 nm whereas the retrieved extinction is at Figure 2 : same global remarks as figure 1 but also : - Why the Raman vertical profile is so different compared to the Klett profile for the opaque cirrus whereas it doesn't for the thin cirrus ? Figure 3 and 4 : please define Raman, Full Res. and iterative and put coherency with elastic and Klett

We took into consideration all the reviewer suggestions. FLG uses aerosol scattering properties from OPAC catalog and cirrus clouds properties from embedded routines. For more information please refer to the provided bibliography. All the figures have been changed as required and now are at high resolution. Thanks for pointing out the discrepancies. Also legend and caption clarity has been improved.

**Comments by Reviewer #3**

The authors have addressed most of the concerns I had before recommending the article for its publication in Atmospheric Measurement Techniques. Personally, I think that the new title proposed is appropriate and that the new study cases for aerosol and cirrus clouds show more convincing results. But I have some minor concern I would like authors address before recommending the final publication: - Although the authors responded well to my concern about the effects of aerosol microphysical properties in aersol radiative forcing computations, I miss such a paragraph in the revised manuscript. - Authors refer in the abstract and in the text to aerosol and clouds optical and geometrical properties. Please, replace by 'aerosol and clouds optical and microphysical properties'. - Although HSRL technique is not used in operational lidar networks such as EARLINET, their potential can not be ignored. Please refer this in the text and add appropriate references. - The statement about the upcoming NASA Aerosol-Clouds-Ecosystems mission is not in the correct place. It is very important to reference such mission, and I recommend to move the sentence at the end of line 141. Also, please update reference to Whiteman et al., 2018: Whiteman, D.N., Pérez-Ramírez, D., Veselovskii, I., Colarco, P., Buchard, V. (2018) Simulations of spaceborne multiwavelength lidar measurements and retrievals of aerosol microphysics. Journal of Quantitative Spectroscopy and Radiative Transfer, 205, 27-39 - Please, define each term of equation 4. - Axis of Figure 1 and Figure 2 are difficult to read

We thank the reviewer for the positive comments. We addressed all the remaining reviewer concerns.

**1 Impact of Varying Lidar Measurement and Data Processing Techniques in evaluating 2 **Cirrus Cloud and Aerosol Direct Radiative Effects.** S. Lolli1,2,1, F. Madonna1, M. Rosoldi1, J. R. Campbell3, E. J Welton4 J. R. Lewis2, Y. 3 4 Gu5, G. Pappalardo1 5 1 
[revised manuscript text omitted]

- 745